# The Contribution of “Individual Participant Data” Meta-Analyses of Psychotherapies for Depression to the Development of Personalized Treatments: A Systematic Review

**DOI:** 10.3390/jpm12010093

**Published:** 2022-01-11

**Authors:** Pim Cuijpers, Marketa Ciharova, Soledad Quero, Clara Miguel, Ellen Driessen, Mathias Harrer, Marianna Purgato, David Ebert, Eirini Karyotaki

**Affiliations:** 1Department of Clinical, Neuro- and Developmental Psychology, Amsterdam Public Health Research Institute, Vrije Universiteit Amsterdam, 1081 HV Amsterdam, The Netherlands; m.ciharova@vu.nl (M.C.); clara.miguelsanz@vu.nl (C.M.); e.karyotaki@vu.nl (E.K.); 2Department of Basic, Clinical Psychology and Psychobiology, Universitat Jaume I, 12006 Castellón, Spain; squero@psb.uji.es; 3CIBER de Fisiopatología de la Obesidad y Nutrición (CIBEROBN), Carlos III Institute of Health, 28029 Madrid, Spain; 4Department of Clinical Psychology, Behavioural Science Institute, Radboud University and Depression Expertise Center, Pro Persona Mental Health Care, 6500 HB Nijmegen, The Netherlands; ellen.driessen@ru.nl; 5Clinical Psychology and Psychotherapy, Friedrich-Alexander-University Erlangen-Nuremberg, 91054 Erlangen, Germany; mathias.harrer@fau.de; 6WHO Collaborating Centre for Research and Training in Mental Health and Service Evaluation, Department of Neurosciences, Biomedicine and Movement Sciences, Section of Psychiatry, University of Verona, 37129 Verona, Italy; marianna.purgato@univr.it; 7Cochrane Global Mental Health, University of Verona, 37129 Verona, Italy; 8Psychology and Digital Mental Health Care, Technical University Munich, 80333 Munich, Germany; david.daniel.ebert@tum.de

**Keywords:** depression, psychotherapy, individual participant data meta-analysis, predictors, moderators

## Abstract

While randomized trials typically lack sufficient statistical power to identify predictors and moderators of outcome, ”individual participant data” (IPD) meta-analyses, which combine primary data of multiple randomized trials, can increase the statistical power to identify predictors and moderators of outcome. We conducted a systematic review of IPD meta-analyses on psychological treatments of depression to provide an overview of predictors and moderators identified. We included 10 (eight pairwise and two network) IPD meta-analyses. Six meta-analyses showed that higher baseline depression severity was associated with better outcomes, and two found that older age was associated with better outcomes. Because power was high in most IPD meta-analyses, non-significant findings are also of interest because they indicate that these variables are probably not relevant as predictors and moderators. We did not find in any IPD meta-analysis that gender, education level, or relationship status were significant predictors or moderators. This review shows that IPD meta-analyses on psychological treatments can identify predictors and moderators of treatment effects and thereby contribute considerably to the development of personalized treatments of depression.

## 1. Introduction

In the past five decades, more than 800 randomized trials have examined the effects of psychotherapies for depression [1]. Several hundreds of trials have compared various psychotherapies, including cognitive behavioral therapy, interpersonal psychotherapy, behavioral activation, problem-solving therapy, psychodynamic therapy, and non-directive counseling, to different types of control conditions [2]. There are also many dozens of trials comparing different types of psychotherapy directly with each other, with pharmacotherapy and with the combination of psychotherapy and pharmacotherapy. This large body of research suggests that all psychotherapies have comparable effects, although non-directive counseling may be somewhat less effective than other therapies [2]. This research has shown that the effects of psychotherapies are comparable to those of pharmacotherapy and that combined treatment is more effective than psychotherapy and pharmacotherapy alone [3], while in the longer term, combined treatment and psychotherapies are more effective than pharmacotherapy [4].

Despite this large body of research, very little is known about who benefits from which treatment. On average most psychotherapies have comparable effects, but it is very well possible that some patients benefit more from one treatment and other patients from another. It has been recognized in the psychotherapy field for a long time that outcome research should not only focus on the effects of treatments but also on “which treatment, by whom, is most effective for this individual with that specific problem and under which set of circumstances” [5]. However, knowledge about predictors and moderators of depression treatment remains scarce.

Predictors can be defined as characteristics that indicate whether a patient benefits from a treatment or not. Specific predictors indicate whether a specific characteristic predicts outcome of therapy compared to a no-treatment control, while non-specific predictors indicate variables that are related to improvement, regardless of comparison or control group (within-group improvement) [6]. Moderators are characteristics that indicate which patients benefit more from one treatment compared to another treatment.

One of the main reasons why predictors and moderators have not been investigated extensively is that very large trials are needed. Most randomized trials are designed to find a significant effect of a treatment at a group-level, and do not have sufficient statistical power to identify predictors or moderators of outcome (individual patient-level). A simulation study has shown that in order to establish a significant moderating variable, the number of included patients has to be increased considerably compared to establishing whether or not an intervention is effective. The required sample size increases exponentially to a factor of more than 100 for more subtle interactions of <20% of the overall effect [7]. This means that a single trial aimed at examining a predictor or moderator should include hundreds of participants per arm to be able to identify this predictor or moderator. Such trials have hardly been carried out in the field of psychotherapy for depression.

One solution to this problem is to increase the sample size of randomized controlled trials of psychotherapies, which would require major funders to acknowledge the problem and the importance of examining predictors and moderators of psychotherapies. The public health importance and the economic costs of depression are substantial enough to justify such studies. However, we do not expect that this will happen soon, considering that more than 800 trials have been conducted in the past five decades, and hardly any of these had sufficient power to examine predictors and moderators.

In this paper, we focus on another approach to examine predictors and moderators of outcome: the “individual participant data” (IPD) meta-analysis. In IPD meta-analyses the primary data of multiple trials are collected, combined into a large merged dataset, and subsequently analyzed jointly. This can be used to examine whether such baseline patient characteristics are associated with the outcomes of therapies, and because the data from multiple trials are combined, the statistical power to examine predictors and moderators of outcome is substantially increased.

IPD meta-analyses are in many ways comparable to conventional pairwise and network meta-analyses, that are based on study-level characteristics extracted from published articles. They require a specific research question, systematic literature searches in bibliographic databases to identify relevant trials, clear eligibility criteria, extraction of study characteristics (e.g., target group, intervention type, setting), and risk of bias assessment of the eligible studies. However, IPD meta-analyses have several important advantages over conventional meta-analyses [8]. For example, they can verify the integrity of included studies, and they result in more statistical power and therefore more precise estimates of the effect size. IPD meta-analyses also allow for standardizing analytic methods across studies (e.g., for handling missing data) and examining rare outcomes such as deterioration and adverse effects. They also make data not published in papers available for analysis. The most important advantage of IPD meta-analyses is probably, however, the increased statistical power that can be used to examine predictors and moderators of outcome. Because of this increased power, IPD meta-analyses are one of the most important tools to develop personalized treatments. We, therefore, conducted a systematic review of IPD meta-analyses of psychological treatments of depression to explore how much they currently have contributed to the knowledge on predictors and moderators of outcome of psychotherapies for depression.

## 2. Methods

### 2.1. Identification and Selection of Studies

The protocol for this meta-analysis was registered at the Open Science Framework on 12 July 2021 (https://osf.io/dkyxt (accessed on 25 November 2021)) [9]. We conducted systematic searches in PubMed and PsycINFO by combining index terms and text words indicative of IPD meta-analyses and depression. The full search string is provided in Appendix A. We also searched the references of included studies. Because several of the authors of the current paper are involved in IPD meta-analyses in this field, we could verify if the searches identified IPD meta-analyses that met our inclusion criteria. The searches were carried out by the first author of the current paper.

We included studies that met the following criteria: (a) IPD meta-analyses of (b) randomized trials comparing the effects of (c) psychological interventions (alone or in combination with pharmacotherapy) for (d) depression (e) in adults to (f) another treatment (psychotherapy or pharmacotherapy) or a control condition. Depression could be defined according to a clinical interview or as a score above a cut-off on a self-rating depression scale. We included pairwise IPD meta-analyses examining one contrast, as well as IPD network meta-analyses examining multiple contrasts simultaneously. IPD meta-analyses were required to identify trials through systematic searches and aiming to contact all authors of identified trials. Thus, IPD meta-analyses that were based on convenience samples of studies without systematic searches were excluded. Unless they reported on additional predictors or moderators, companion papers based on the same sample of studies were excluded. Depression could be established in the trials with a diagnostic interview or with a score above a cut-off on a self-report measure. No language restrictions were applied.

### 2.2. Data Extraction

We extracted the following characteristics of the included IPD (network) meta-analyses: the comparisons examined, the number of included trials, the proportion of included studies for which IPD were retrieved, and the total number of participants included in the IPD meta-analyses. We also extracted the overall effect size for the contrast(s) examined as well as the significant predictors and moderators that were identified in the study. Because the power in IPD meta-analyses is high, non-significant predictors and moderators probably indicate that these variables are not relevant as predictors or moderators. We, therefore, also listed the predictors and moderators that were not found to be significantly associated with the outcome. We distinguished between specific predictors (a characteristic predicting outcome of therapy compared to a no-treatment control), non-specific predictors (variables that are related to improvement, regardless of comparison or control groups; within-group improvement), and moderators (characteristics that indicate which patients benefit more from one treatment compared to another treatment).

The quality of the included meta-analyses was assessed with the AMSTAR-2 (“MeaSurement Tool to Assess systematic Reviews”), a critical appraisal tool for systematic reviews [10]. AMSTAR-2 assesses several core characteristics of systematic reviews, including the use of participants, intervention, comparator, outcome (PICO) in formulating the research question, whether the methods were established before the review was conducted, an explanation for the selection of the study design to be included, the comprehensiveness of the search strategy, study selection by at least two reviewers, data extraction by at least two reviewers, providing a list of excluded studies with reasons, a detailed description of included studies, assessment of risk of bias in included studies, reporting sources of funding for the included studies, appropriate methods for pooling results of individual studies, assessment of the impact of risk of bias on the outcomes, a discussion of the impact of risk of bias on results, an explanation and discussion of heterogeneity, assessment of publication bias, and reporting potential conflict of interest and funding for the review. Each item was rated as positive (yes), probably positive (partial yes), or negative (no). All assessments of these criteria were conducted by two independent researchers, and disagreements were solved by discussion or, when needed, were discussed with a third reviewer.

Because several authors of the current paper were involved in the majority of included IPD meta-analyses, the quality assessments were conducted by two authors who were not involved in any of the included IPD meta-analyses (M.C. and S.Q.).

### 2.3. Integration of Findings

We created an overview of the identified predictors and moderators for each IPD meta-analysis. We integrated these results in a narrative review, focusing on the type of predictors and moderators and the significant and non-significant predictors and moderators.

## 3. Results

### 3.1. Selection and Inclusion of Studies

After examining a total of 361 abstracts (209 after removal of duplicates), we retrieved 22 full-text papers for further consideration. We excluded 12 of the retrieved papers. The PRISMA (Preferred Reporting Items for Systematic Reviews) flowchart describing the inclusion process, including the reasons for exclusion, is presented in Figure 1. A total of 10 IPD meta-analyses met the inclusion criteria for this review [11,12,13,14,15,16,17,18,19,20].

### 3.2. Characteristics of the Included Studies

A summary of key characteristics of the included IPD meta-analyses is presented in Table 1. Eight of the 10 included studies were pairwise IPD meta-analyses examining one treatment contrast, while two were IPD network meta-analyses. The number of included trials ranged from 3 to 39 (median: 12), and the number of included patients ranged from 482 to 8107 (median: 1943). The percentage of trials that was included in the IPD meta-analysis of the total number of trials identified through the searches ranged from 55.2% to 100% (all but two included more than 80%, and three included 100% of the identified trials).

Of the eight pairwise IPD meta-analyses, six compared psychological interventions to control groups (usual care, waitlist, pill placebo, other inactive control,), one compared psychological interventions to antidepressant medications (ADMs), and one compared combined treatment to ADM. Five of the eight examined cognitive behavioral therapy (CBT): one low-intensity CBT [11], two internet-based CBT (iCBT) [15,16], and two face-to-face CBT [12,14,21]. The other three examined internet-based indicated prevention programs [13], mindfulness-based CBT [17], and psychodynamic therapy [18]. One of the two meta-analyses on face-to-face CBT was examined in two separate papers using the same data, one examining gender and other sociodemographic characteristics [21] and one examining baseline severity [12]. One of the two IPD network meta-analyses compared Cognitive Behavioral Analysis System of Psychotherapy (CBASP) to ADM or combined treatment of CBASP and ADM [19], while the other compared guided iCBT with unguided iCBT and control groups [20].

In the current systematic review we focus on the outcomes of the IPD (network) meta-analyses on predictors and moderators, but for most IPD meta-analyses other outcomes were also reported, typically in other papers. These other papers focused for example on negative outcomes [22,23,24] but also on other outcomes such as cost-effectiveness [25], symptom-specific outcomes (network analyses) [26], and IPD ”component” network meta-analyses [27]. In this study we focus on the main outcomes of the IPD meta-analyses on depression severity and on predictors and moderators of these main outcomes.

### 3.3. AMSTAR-2 Ratings

The AMSTAR-2 ratings for each of the 10 included meta-analyses are presented in Table 1, and the aggregated ratings across all reviews are reported in Figure 2. The specification of PICO was judged as adequate for all 10 studies. The majority of the meta-analyses did not register a protocol (*n* = 4; 40%). They all gave a good justification for the selection of study designs. Comprehensiveness of the literature search was rated as “partial yes” in 9 of the 10 meta-analyses. Two reviewers independently selected studies and extracted data in three meta-analyses. A list of excluded studies with reasons was not provided in any meta-analysis. The description of included studies was rated positive or probably positive in all meta-analyses. Risk of bias was rated as positive or probably positive in nine studies. Sources of funding for included studies was reported in only one meta-analysis. Statistical combination of results was rated as positive in 8 of the 10 meta-analyses. All 10 meta-analyses examined the impact of risk of bias on the results and accounted for risk of bias when interpreting the results. Heterogeneity was examined in seven meta-analyses. Publication bias was also examined in seven meta-analyses. Conflict of interest was reported in nine meta-analyses.

None of the studies scored positive (“yes” or “probably yes”) on all 16 items of AMSTAR-2, 5 scored positive on 12 or 13 items, while the other 5 were rated positive on 8 to 11 items.

### 3.4. Predictors and Moderators in Pairwise IPD Meta-Analyses

Table 2 summarizes the effect sizes found for all 10 IPD (network) meta-analyses as well as the significant and non-significant predictors and moderators. The IPD meta-analysis from Bower and colleagues found that baseline severity of depression was a significant specific predictor of outcome (higher severity was associated with better outcomes) in low-intensity CBT versus control conditions [11]. Another meta-analysis found that baseline severity was not a significant moderator of outcome for CBT and ADM for depression [12]. Using the same data, Cuijpers and colleagues found that gender was not a significant predictor (specific and non-specific) or moderator in studies comparing CBT, ADM and pill placebo [21]. The IPD meta-analysis by Furukawa and colleagues focusing on trials comparing CBT with pill placebo found no association between baseline severity and outcome in a sample of studies comparing CBT with pill placebo [14].

One IPD meta-analysis focused on specific predictors of outcome of guided iCBT compared to control groups [15]. It was found that older adults, people who were native-born, and people who had higher baseline severity benefitted more from guided iCBT. Gender, relationship status, education, medication use, anxiety, previous depressive episodes, and alcohol problems were not found to be predictors of outcome. Another IPD meta-analysis focused on unguided iCBT compared to control groups [16]. Age, gender, education, relationship status, anxiety, and baseline severity were examined, but none of these variables was found to be a significant predictor of outcome.

One IPD meta-analysis focusing on mindfulness-based CBT for relapse prevention versus control groups found that baseline severity was a significant specific predictor of outcome (better outcomes for more severe depression), but age, gender, education, and relationship status were not significant predictors [17]. Another IPD meta-analysis focused on trials comparing combined treatment of psychodynamic treatment and pharmacotherapy versus pharmacotherapy alone, but did not report on predictors or moderators [18]. The IPD meta-analysis published by Reins and colleagues examined internet-based interventions aimed at subthreshold depression compared with control groups [13]. Higher baseline severity and older age were found to be associated with better outcomes, while gender, relationship status, employment status, previous therapy, medication use, anxiety, medical condition, and level of education were not associated with outcome.

### 3.5. Predictors and Moderators in IPD Network Meta-Analyses

Two IPD network meta-analyses were included in this systematic review. In the first one CBASP was compared with ADM and combined treatment of CBASP and ADM [19]. In the second one, guided and unguided internet-based interventions were compared with each other and with control groups [20]. Both network IPD meta-analyses used a different approach from the IPD meta-analyses described before. Models were developed in which available predictors, moderators, and selected interactions between these were combined to generate outcome predictions for a specific participant. These models can predict the outcome for one specific participant, based on the available characteristics in the study. For both studies interactive websites (shiny apps) are available where the characteristics of a participant can be entered to generate outcome results for each intervention or control group (links are given in Table 2).

Examined moderators and predictors are reported in Table 2. The most important moderators and predictors for the IPD network meta-analysis of CBASP were baseline depression, anxiety, prior pharmacotherapy, age, and subtypes of chronic depression [19]. The most important moderator in the IPD network meta-analysis on guided and unguided iCBT was baseline severity. Among participants with mild depression no significant difference was found between guided and unguided iCBT, which both were superior to control conditions. Among participants with more severe depression, guided iCBT was superior to unguided iCBT [20].

## 4. Discussion

We conducted a systematic review to provide an overview of predictors and moderators for psychological interventions for depression as identified in IPD (network) meta-analyses. In the 10 included IPD meta-analyses, we found several significant predictors and moderators. Although baseline severity was not found to be a significant predictor or moderator in two IPD meta-analyses, six did find that psychological interventions were more effective when baseline severity was higher. Age was also found to be a significant predictor in two meta-analyses, but not in two others. Because power was high in most IPD meta-analyses, non-significant findings are also relevant because they indicate that these variables are probably not particularly relevant as predictors and moderators. We did not find in any IPD meta-analysis that gender, education level, or relationship status were significant predictors or moderators.

These findings do suggest that psychological interventions may not be equally effective across different target groups. That is important for the further development of personalized treatments of depression because it supports the assumption that such personalized treatment may be more effective than standard treatments.

The IPD network meta-analyses that were included in this systematic review took a further step towards personalized treatment selection. They included all the available information on all variables to create prediction models for individual patients with specific characteristics. This goes beyond the examination of whether or not a variable is a significant predictor or moderator by predicting how one individual will benefit from a specific treatment.

All IPD meta-analyses included suffered from the problem that only a limited number of variables were available as potential predictor or moderator. The trials that were included in the meta-analyses used many different predictors and moderators, limiting the potential of the IPD meta-analyses to include a broad set of variables. Current research on psychological interventions would be considerably strengthened if a common set of outcome measures, predictors, and moderators was included across all new randomized trials. It is suspected that variables such as comorbid personality problems, comorbid substance use problems, childhood adversities, traumas, the duration of depressive symptoms, the number of previous episodes and available coping resources are important candidates for significant predictors and moderators [28,29]. However, as long as these variables are not included in randomized trials, IPD meta-analyses will not be able to examine whether these are indeed significant predictors and moderators.

This study has some clinical implications. The IPD network meta-analyses have resulted in findings that can predict outcomes for specific individuals with specific characteristics, although these should be considered with caution because of the indirect evidence. It does seem that severity may be an indicator of better outcomes for several interventions, and there is also evidence that CBT is as effective as antidepressants in those with more severe depression. It would be good, however, to validate such findings in new randomized trials.

We have to keep in mind that significant predictors and moderators that are identified in IPD meta-analyses are only correlational associations. New randomized trials are needed to confirm that predictors and moderators can indeed improve outcomes for specific groups of patients. Such randomized trials should preferably create decision models based on the available evidence for assigning a patient to an intervention, which then is compared with a group of patients who are not assigned to one specific intervention. Similarly, the prediction models developed in the network IPD meta-analyses need external validation before they can be used to guide treatment selection. Before these models can be used in clinical practice they will also have to take patient preferences and needs into account. These next step has not yet been taken in the field of psychotherapy for depression but is needed to bring the findings of these studies one step further to clinical practice.

Probably the most important contribution of IPD meta-analyses is that they can examine significant and non-significant predictors and moderators with sufficient statistical power. However, we also found several other types of studies that used the methodology of IPD meta-analyses, including studies on deterioration and negative effects [22,23,24], economic outcomes, and networks of symptom outcomes [26]. These studies show that IPD meta-analyses are an important new method to examine the outcomes of treatments.

This study has several important strengths. To the best of our knowledge, it is the first systematic review of IPD meta-analyses of psychological interventions for depression. It also shows that these meta-analyses have the potential to contribute considerably to the further development of personalized treatments, especially when new trials include a common set of potentially important predictors and moderators. There are, however, also several limitations that have to be acknowledged. First, the number of included IPD meta-analyses was relatively small, and there was some overlap among the included studies. It is beyond the scope of the current paper to look at this overlap, but this may have influenced the outcomes on predictors and moderators.

Furthermore, the quality of the included meta-analyses was not optimal. We also only focused on predictors and moderators at post-test, while in the end the longer-term effects of treatments are at least as relevant as the short-term outcomes. We did not find that pharmacotherapy was a significant predictor or moderator, but that may have been related to the fact that most IPD meta-analyses did not examine this.

IPD meta-analyses in general also have several important disadvantages and challenges. They are much more resource intensive than conventional meta-analyses, they rely on the willingness of researchers to contribute the data of trials, studies may not be able to contribute data and result in selection bias, outcome measures may vary too much, and they cannot fix problems of bias related to the study. They also can only be used to examine predictors and moderators that are available in the included trials.

Despite these limitations we can conclude that IPD meta-analyses on psychological treatments have been able to identify predictors and moderators of treatment outcome, and thereby can probably contribute considerably to the development of personalized treatments of depression.

## Figures and Tables

**Figure 1 jpm-12-00093-f001:**
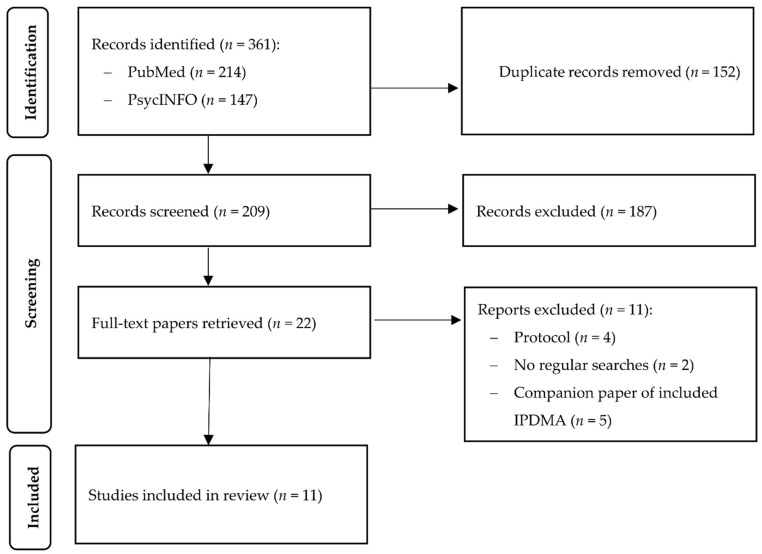
Flowchart for the inclusion of studies.

**Figure 2 jpm-12-00093-f002:**
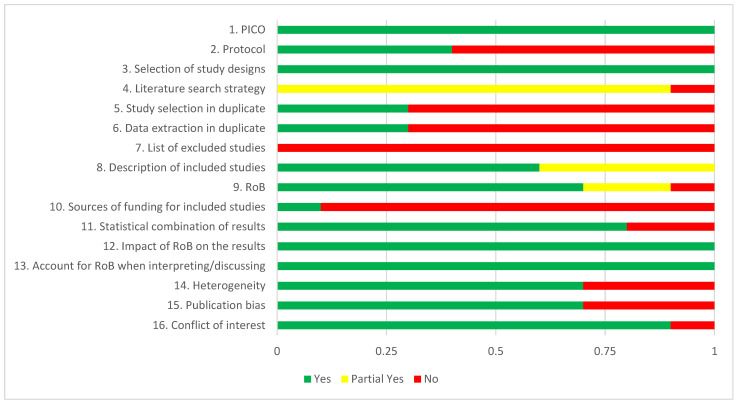
AMSTAR-2 ratings of included studies. Abbreviations: PICO: Participants, Intervention, Comparator, Outcome; RoB: risk of bias.

**Table 1 jpm-12-00093-t001:** Selected characteristics of IPD meta-analyses and IPD network meta-analyses on psychotherapies for adult depression.

							AMSTAR-2 ^a)^																
Study	Intervention	Comparison	Type ^b)^	N_st_	N_part_	Proportion	1	2	3	4	5	6	7	8	9	10	11	12	13	14	15	16	Tot Y/PY
Bower et al., 2013 [11]	Low intens. CBT	Usual care	ma	16	2470	55.2 (16/29)	Y	N	Y	N	N	Y	N	PY	N	N	Y	Y	Y	N	Y	Y	9
Weitz et al., 2015 [12]	CBT	ADM	ma	16	1700	66.7 (16/24)	Y	Y	Y	PY	Y	N	N	Y	Y	N	Y	Y	Y	Y	N	Y	12
Reins et al., 2020 [13]	Indicated prevention	Usual care or WL	ma	7	2186	87.5 (7/8)	Y	N	Y	PY	N	N	N	Y	Y	N	N	Y	Y	N	N	Y	8
Furukawa et al., 2017 [14]	CBT	Pill placebo	ma	5	509	100 (5/5)	Y	Y	Y	PY	N	N	N	PY	Y	Y	Y	Y	Y	Y	Y	Y	13
Karyotaki et al., 2018 [15]	Guided iCBT	Any control	ma	24	4889	88.9 (24/27)	Y	N	Y	PY	Y	N	N	Y	Y	N	Y	Y	Y	Y	Y	Y	12
Karyotaki et al., 2017 [16]	Unguided iCBT	Any control	ma	13	3876	81.3 (13/16)	Y	N	Y	PY	Y	Y	N	Y	Y	N	Y	Y	Y	Y	Y	N	12
Kuyken et al., 2016 [17]	MBCT	Any control or active treatment	ma	10	1258	90.0 (9/10)	Y	Y	Y	PY	N	Y	N	Y	Y	N	Y	Y	Y	Y	Y	Y	13
Driessen et al., 2020 [18]	Dynamic+ADM	ADM	ma	7	482	100 (7/7)	Y	N	Y	PY	N	N	N	PY	Y	N	Y	Y	Y	Y	Y	Y	11
Furukawa et al., 2018 [19]	CBASP vs. ADM	vs. COMB	nma	3	1036	100 (3/3)	Y	Y	Y	PY	N	N	N	Y	PY	N	N	Y	Y	N	N	Y	9
Karyotaki et al., 2021 [20]	Guided iCBT vs. Unguided iCBT	vs. any control	nma	39	8107	92.9 (39/42)	Y	N	Y	PY	N	N	N	PY	PY	N	Y	Y	Y	Y	Y	Y	11

^a)^ The AMSTAR-2 items refer to 1. the use of participants, intervention, comparator, outcome (PICO) in formulating the research question, 2. whether the methods were established before the review was conducted, 3. an explanation for the selection of the study design to be included, 4. the comprehensiveness of the search strategy, 5. study selection by at least two reviewers, 6. data extraction by at least two reviewers, 7. providing a list of excluded studies with reasons, 8. a detailed description of included studies, 9. assessment of risk of bias in included studies, 10. reporting sources of funding for the included studies, 11. appropriate methods for pooling results of individual studies, 12. assessment of the impact of risk of bias on the outcomes, 13. a discussion of the impact of risk of bias on results, 14. an explanation and discussion of heterogeneity, 15. assessment of publication bias, 16. reporting potential conflict of interest and funding for the review. ^b)^ Type of IPD meta-analysis (ma = pairwise IPD meta-analysis; nma = IPD network meta-analysis). Abbreviations: ADM: antidepressant medication; CBASP: Cognitive Behavioral Analysis System of Psychotherapy; CBT: cognitive behavioral therapy; COMB: combined treatment; Dynamic: psychodynamic psychotherapy; iCBT: internet-based CBT; Low intens. CBT: low-intensity CBT; MBCT: mindfulness-based CBT; N_part_: number of participants; N_st_: number of studies; Tot Y/PY: total number of “Yes” on all AMSTAR-2 domains; vs: versus; WL: waiting list; N: No; PY: Probably yes.

**Table 2 jpm-12-00093-t002:** Effect sizes and predictors identified in IPD meta-analyses of psychotherapies for adult depression.

Study	Contrast ^a)^	Effect Size	Significant Predictors/Moderators ^b)^	Non-significant Predictors/Moderators ^c)^
	IPD meta-analyses			
Bower et al., 2013 [11]	Low-intensity CBT vs. CTR	CBT > CTR: SMD = −0.42 (95% CI: −0.55; −0.29)	SPE: Baseline severity	-
Weitz et al., 2015 [12]	CBT vs. ADM	ADM > CBT (on HAM-D: β = −0.88; *p* = 0.03)	-	SPE/NSP/MOD: Gender MOD: Baseline severity
Furukawa et al., 2017 [14]	CBT vs. pill placebo	CBT > placebo: SMD: −0.22 (95% CI: −0.42; 0.02)	-	SPE: Baseline severity
Karyotaki et al., 2018 [18]	Guided iCBT vs. CTR	Guided iCBT > CTR; OR = 2.49 response; OR = 2.41 remission	SPE: Older age; native-born; baseline severity	SPE: Sex; relationship; education; medication use; anxiety; previous episodes; alcohol problems
Karyotaki et al., 2017 [16]	Unguided iCBT vs. CTR	Unguided iCBT > control; g = 0.27	SPE: None	SPE: Age, sex, education, relation-ship status anxiety, baseline severity
Kuyken et al., 2016 [17]	MBCT for relapse vs. CTR	MBCT > CTR: HR of relapse = 0.69 (95% CI: 0.58; 0.82)	SPE: Baseline severity	SPE: Age, sex, education, relationship status
Driessen et al., 2020 [18]	Dynamic vs. combined treatment	Combined > dynamic therapy; d = 0.26	-	-
Reins et al., 2020 [13]	Internet interventions for subthreshold depression	Internet interventions > control; d = 0.39	SPE: Higher baseline severity; Older age	SPE: Gender; relationship; employment; previous therapy; medication use; anxiety; medical condition; education
	IPD network meta-analyses		Examined moderators/predictors and models	
Furukawa et al., 2018 [19]	CBASP for persistent depression	COMB > CBASP: 2.9 (1.3; 4.6) HAMD points COMB > ADM: 2.9 (1.6; 4.3) HAMD points CBASP = ADM: 0.1 (−1.6; 1.7) HAMD points	Examined moderators/predictors: Baseline depression, anxiety, prior medication, childhood neglect, and several interactions Outcomes: Baseline depression, anxiety, prior pharmacotherapy, age, and depression subtypes moderated their relative efficacy. URL: https://kokoro.med.kyoto-u.ac.jp/CBASP/prediction/ (accessed on 25 November 2021)	
Karyotaki et al., 2021 [20]	Guided and unguided iCBT	g-iCBT > u-iCBT: −0.8 (−1.4; −0.2) PHQ-9 points g-iCBT > TAU: −1.7 (−2.3; −1.1) PHQ-9 points g-iCBT > WL: −3.3 (−3.9; −2.6) PHQ-9 points u-iCBT > TAU: −0.9 (−1.5; −0.3) PHQ-9 points u-iCBT > WL: −2.5 (−3.2; −1.8) PHQ-9 points	Examined moderators/predictors: Baseline severity, age, sex, educational level, relationship status, employment status, treatment adherence. Outcomes: Baseline severity was the most important moderator. In mild depression differences between unguided and guided iCBT were small; in more severe depression guided iCBT is more effective. URL: https://cinema.ispm.unibe.ch/shinies/iCBT/ (accessed on 25 November 2021)	

^a)^ Extensive definitions of the different types of therapies can be found in Cuijpers et al., 2020. ^b)^ Predictors defined as characteristics that indicate whether a patient benefits from a treatment or not. Specific predictors (SPEs) indicate whether a specific characteristic predicts outcome of therapy compared to a no-treatment control, while non-specific predictors (NPEs) indicate variables that are related to improvement, regardless of comparison or control group (within-group improvement). Moderators (MODs) are characteristics that indicate which patients benefit more from one treatment compared to another treatment. ^c)^ Because these studies are based on large sample sizes, non-significant variables are also important because they are probably not associated with the outcome. Abbreviations: ADM: antidepressant medication; CBASP: Cognitive Behavioral Analysis System of Psychotherapy; CBT: cognitive behavioral therapy; CTR: control group; HAM-D: Hamilton Rating scale for depression; HR: hazard ratio; iCBT: internet-based CBT; IPD: individual participant data; MBCT: mindfulness-based CBT; MOD: moderator; NSP: non-specific predictor; OR: odds ratio; PHQ-9: Patient Health Questionnaire-9 items; SMD: standardized mean difference; SPE: specific predictor; TAU: treatment as usual; u-iCBT: unguided iCBT; WL: waiting list.

## Data Availability

All data used are reported in the paper or can be found in the included papers.

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
