# Peer review of "The Contribution of “Individual Participant Data” Meta-Analyses of Psychotherapies for Depression to the Development of Personalized Treatments: A Systematic Review"

_jpm, 2022, doi:10.3390/jpm12010093_

Round 1

Reviewer 1 Report

Comment to the authors:

In their work, Cuijpers and colleagues conducted a systematic review on the contribution of IPDMAs of psychotherapies for depression on research on and development of personalized treatments. The Authors conclude that personalized treatments may be more effective than standard treatments.

Considering long-lasting research on the overall effectiveness of psychotherapy for depression and treatment personalization, the review addresses a very important and relevant question in the field. A strength of the manuscript is the clarity and conciseness in which the theoretical background, the methods, and results are described. Nevertheless, before accepting this manuscript for publication, I would recommend to revise some minor aspects.

Methods:

I really enjoyed reading the methods section. It was concisely written and well structured. I particularly liked the idea to split pairwise IPD from IPD network analysis. At some points, I had a hard time reading table 2. The authors might consider restructuring it in a way that makes it more comprehensible, particularly the column of non-significant predictors and moderators and the lower part of the table on IPD network MAs.

Discussion:

For good reasons, the authors mention the fact, that there was a considerable overlap of included original studies in the different meta-analyses (357). This could result in an overrepresentation and dominance of some original trials and their individual patient data. The authors should give detailed information on which trials were included in more than one of the selected IPDMAs and how this might affect results and finally, the conclusions of their review.

Considering the findings of the pairwise IPDMAs on high baseline severity and to some degree older age being the only significant (among many) predictors/moderators, the drawn conclusion that “psychological interventions are not equally effective across different target groups” (311) should be attenuated.

Additionally, the reader might be interested in a more detailed discussion, interpretation, and theoretical reasoning of these findings (and non-findings).

I was wondering if the authors could elaborate and outline the clinical implications of their findings more clearly in the discussion section.

In the end, I`d like to add that I strongly support the idea of establishing a core set of measures, predictors, and moderators to be included in future RCTs on treatments for depression.

Author Response

Comment:   In their work, Cuijpers and colleagues conducted a systematic review on the contribution of IPDMAs of psychotherapies for depression on research on and development of personalized treatments. The Authors conclude that personalized treatments may be more effective than standard treatments.

                  Considering long-lasting research on the overall effectiveness of psychotherapy for depression and treatment personalization, the review addresses a very important and relevant question in the field. A strength of the manuscript is the clarity and conciseness in which the theoretical background, the methods, and results are described. Nevertheless, before accepting this manuscript for publication, I would recommend to revise some minor aspects.

Reply:         We thank the reviewer for these positive comments and we hope that we revised the paper sufficiently to solve the reported minor issues.

Comment:   Methods: I really enjoyed reading the methods section. It was concisely written and well structured. I particularly liked the idea to split pairwise IPD from IPD network analysis. At some points, I had a hard time reading table 2. The authors might consider restructuring it in a way that makes it more comprehensible, particularly the column of non-significant predictors and moderators and the lower part of the table on IPD network MAs.

Reply:         We have added two footnotes to Table 2 to make it easier to understand, one to explain how predictors and moderators are defined and one to explain why non-significant variables are important:

                  “b) Predictors defined as characteristics that indicate whether a patient benefits from a treatment or not. Specific predic-tors (SPE) indicate whether a specific characteristic predicts outcome of therapy compared to a no-treatment control, while non-specific predictors (NPE) indicate variables that are related to improvement, regardless of comparison or control group (within-group improvement). Moderators (MOD) are characteristics that indicate which patients benefit more from one treatment compared to another treatment.

  1. c) Because these studies are based on large sample sizes, non-significant variables are also important because they are proba-bly not associated with the outcome”

                  We did not change the table on the IPD network meta-analyses. We could have made a different table of that, but that would also be confusing.

Comment:   Discussion: For good reasons, the authors mention the fact, that there was a considerable overlap of included original studies in the different meta-analyses (357). This could result in an overrepresentation and dominance of some original trials and their individual patient data. The authors should give detailed information on which trials were included in more than one of the selected IPDMAs and how this might affect results and finally, the conclusions of their review.

Reply:         This is a good point. However, it is beyond the scope of the current paper to make a complete overview of all included primary studies and it cannot be estimated how this may have influenced the outcomes. We have mentioned this more extensively as a limitation of the paper in the Discussion section: “First, the number of included IPD meta-analyses was relatively small, and there was some overlap among the included studies. It is beyond the scope of the current paper to look at this overlap, but this may have influenced the outcomes on predictors and moderators.”

Comment:   Considering the findings of the pairwise IPDMAs on high baseline severity and to some degree older age being the only significant (among many) predictors/moderators, the drawn conclusion that “psychological interventions are not equally effective across different target groups” (311) should be attenuated.

Reply:         We have formulated this more cautiously in the revised paper: “These findings do suggest that psychological interventions may not be equally ef-fective across different target groups.”

Comment:   Additionally, the reader might be interested in a more detailed discussion, interpretation, and theoretical reasoning of these findings (and non-findings).

Reply:         We thank the reviewer for this comments. However, we already have a long Discussion section and it is not clear what else the reviewer would like to see added.

Comment:   I was wondering if the authors could elaborate and outline the clinical implications of their findings more clearly in the discussion section.

Reply:         We have added a paragraph on the clinical implications to the Discussion section: “This study has some clinical implications. The IPD network meta-analyses have resulted in findings that can predict outcomes for specific individuals with specific characteristics, although these should be considered with caution because of the indirect evidence. It does seem that severity may be an indicator of better outcomes for several interventions, and there is also evidence that CBT is as effective as antidepressants in those with more severe depression. It would be good, however, to validate such findings in new randomised trials.”

Comment:   In the end, I`d like to add that I strongly support the idea of establishing a core set of measures, predictors, and moderators to be included in future RCTs on treatments for depression.

Reply:         We thank the reviewer for this support. We assume that no revision is needed because this point is discussed extensively in the text (lines 328-339).

Reviewer 2 Report

It is a remarkable paper about the psychotherapies for depression to the development of personalized treatments. 

  1. The potential influence of pharmacotherapy on the results should be properly adjusted.
  2. In method section, an operational definition of depression should be more strictly described.
  3. In method section, the psychological interventions should be more definitely described (i.e., psychodynamic psychotherapy, supportive psychotherapy, interpersonal therapy, cognitive-behavioral therapy, and others)

Author Response

Reviewer 2

Comment:   It is a remarkable paper about the psychotherapies for depression to the development of personalized treatments.

Reply:         We thank the reviewer for these positive comments.

Comment:   The potential influence of pharmacotherapy on the results should be properly adjusted.

Reply:         We added the following limitation to the Discussion section: “We did not find that pharmacotherapy was a significant predictor or moderator, but that may have been related to the fact that most IPD meta-analyses did not examine this.”

Comment:   In method section, an operational definition of depression should be more strictly described.

Reply:         We added a definition of depression to the methods section: “Depression could be defined according to a clinical interview or as a score above a cut-off on a selfrating depression scale.”

Comment:   In method section, the psychological interventions should be more definitely described (i.e., psychodynamic psychotherapy, supportive psychotherapy, interpersonal therapy, cognitive-behavioral therapy, and others)

Reply:         We have added a footnote to Table 2 (in which the different therapies are described most extensively) and referred to a paper we wrote with extensive definitions of the main types of therapy for depression: “Extensive definitions of the different types of therapies can be found in Cuijpers et al., 2020[29]”